# Skin-Inspired Magnetoresistive Tactile Sensor for Force Characterization in Distributed Areas

**DOI:** 10.3390/s25123724

**Published:** 2025-06-13

**Authors:** Francisco Mêda, Fabian Näf, Tiago P. Fernandes, Alexandre Bernardino, Lorenzo Jamone, Gonçalo Tavares, Susana Cardoso

**Affiliations:** 1INESC Microsistemas e Nanotecnologias (INESC-MN), 1000-029 Lisbon, Portugal; fabian.naf@tecnico.ulisboa.pt (F.N.); tiago.fernandes@inesc-mn.pt (T.P.F.); scardoso@inesc-mn.pt (S.C.); 2Instituto Superior Técnico (IST), Universidade de Lisboa, 1049-001 Lisbon, Portugal; alexandre.bernardino@tecnico.ulisboa.pt (A.B.); goncalo.tavares@tecnico.ulisboa.pt (G.T.); 3INESC Investigação e Desenvolvimento (INESC ID), 1000-029 Lisbon, Portugal; 4Instituto de Sistemas e Robótica (ISR), 1049-001 Lisbon, Portugal; 5Computer Science Department, University College London (UCL), London WC1E 6BT, UK; l.jamone@ucl.ac.uk

**Keywords:** tactile sensor, artificial skin, magnetoresistive sensor, magnetorheological elastomer, neural network, instrumentation

## Abstract

Touch is a crucial sense for advanced organisms, particularly humans, as it provides essential information about the shape, size, and texture of contacting objects. In robotics and automation, the integration of tactile sensors has become increasingly relevant, enabling devices to properly interact with their environment. This study aimed to develop a biomimetic, skin-inspired tactile sensor device capable of sensing applied force, characterizing it in three dimensions, and determining the point of application. The device was designed as a 4 × 4 matrix of tunneling magnetoresistive sensors, which provide a higher sensitivity in comparison to the ones based on the Hall effect, the current standard in tactile sensors. These detect magnetic field changes along a single axis, wire-bonded to a PCB and encapsulated in epoxy. This sensing array detects the magnetic field from an overlayed magnetorheological elastomer composed of Ecoflex and 5 µm neodymium–iron–boron ferromagnetic particles. Structural integrity tests showed that the device could withstand forces above 100 N, with an epoxy coverage of 0.12 mL per sensor chip. A 3D movement stage equipped with an indenting tip and force sensor was used to collect device data, which was then used to train neural network models to predict the contact location and 3D magnitude of the applied force. The magnitude-sensing model was trained on 31,260 data points, being able to accurately characterize force with a mean absolute error ranging between 0.07 and 0.17 N. The spatial sensitivity model was trained on 171,008 points and achieved a mean absolute error of 0.26 mm when predicting the location of applied force within a sensitive area of 25.5 mm × 25.5 mm using sensors spaced 4.5 mm apart. For points outside the testing range, the mean absolute error was 0.63 mm.

## 1. Introduction

All living organisms depend on understanding their surroundings to interact with the world. As humans, we depend on our senses to gather this information. Touch is particularly crucial for a wide range of tasks, providing a feel for the objects with which we interact. Shape, size, temperature, and texture are all provided, in some form, by touch, and this information allows us to handle them accordingly [1,2,3,4].

In recent years, with robotics, automation, and artificial intelligence occupying a center stage in engineering development, human-mimicking devices are more relevant than ever [5,6,7]. Consequently, it is reasonable to consider that a skin-inspired tactile sensor could be the next step in this field of robotics [7,8,9,10,11,12,13,14,15,16,17,18]. Such a bio-inspired device should be able to sense and characterize its surroundings with the added benefits of biological skin. This study employed some of these characteristics [5], aiming to develop an accurate and resistant tactile sensor based on a flexible and biocompatible artificial skin.

The interest in tactile sensing devices coincides with the rise in robotics, and their development can be traced back to the last 40 years. With the inception of the first piezoelectric sensing arrays, Kinoshita et al. [19] created a robot hand with hybrid visual–tactile sensing. The implementation of flexible materials in tactile sensors [20] allowed for more accurately mimicking biological skin, which would be further expanded with the turn of the century [21]. These discoveries have allowed us to increase the complexity of tactile sensing for new purposes, such as the perception of texture, shape, and hardness of objects.

Tactile sensors can be classified based on their underlying physical principles [11,21]. These include capacitive [22,23], piezoelectric [24,25], piezoresistive [26,27], optical [28,29,30,31], fiber optic [32,33], electrical impedance tomography [34,35], and magnetic sensors [36,37,38,39,40], each offering distinct advantages depending on the application. Selecting a sensor requires careful consideration of several key criteria, including spatial resolution, sensitivity, frequency response, hysteresis, size, flexibility, mechanical properties, and overall robustness [41]. Each of these factors plays a crucial role in determining the sensor’s performance and suitability for a given application. Magnet-based tactile sensors are particularly suitable for a biomimetic approach due to their ability to characterize the applied stimulus in more than one direction [42], the potential to employ flexible materials, and cost-effectiveness. Currently, the most commonly used magnetic sensors for tactile sensing applications are Hall effect-based (or Hall sensors), for their low cost, three-dimensional sensitivity, and low volume. In recent years, other promising alternatives have been investigated, namely thin film magnetoresisitve sensors—giant (GMR) and tunneling magnetoresistive (TMR)—due to their comparatively higher sensitivity [39]. The novelty of this paper lies in the use of TMR sensors [43] for a tactile sensor device, which should provide a higher sensitivity compared to other reported works.

As mentioned above, the film/skin approach to tactile sensors is advantageous, not only for prosthetic and automation purposes, but also for applications that benefit from a large sensing area. In addition, it is a relatively compact architecture that also provides flexibility. With this in mind, two of these artificial skin devices have been produced in the past few years: uSkin [44] and ReSkin [45].

The uSkin device is a commercial product developed by XELA Robotics based on the design described in [46,47], which consists of a matrix-like array of 3-axis Hall-effect sensors covered by a membrane of elastomeric material (Ecoflex) with embedded permanent magnets, each magnet centered on top of each Hall-effect sensor. When pressure is applied to the external surface of the membrane, the magnets shift from their initial position, and this motion is picked up by the Hall-effect sensors. This data is then interpreted to obtain information about the intensity, direction, and point of application of the force. Among the various products developed by the company, the one most comparable to the present study demonstrates a sensitivity of up to 0.70 mN [48].

ReSkin is a prototype device that comprises a magnetized soft material placed over a flexible Hall-effect-based sensing component that is able, through machine learning techniques, to sense both normal and shear forces and estimate the contact points in which they originate. Regarding the characterization of applied stimuli, ReSkin exhibited a mean squared error (MSE) of 0.142 N^2^ for force measurements and 0.514 mm^2^ for location predictions in a 20 mm × 20 mm [45] area.

Other recent studies [49,50,51,52] have also employed the approach of Reskin, using magnetorheological elastomer and an array of Melexis 3D Hall effect sensors [53]. The earliest of these [49] reported a force perception with a mean error from 0.25 N to 0.44 N and the ability to distinguish between 25 distinct locations in a 15 mm^2^ area with an accuracy of over 98% [49]. The technology developed by Fang et al. exhibits a force perception characterized by an MSE of 0.04 N^2^ and can distinguish stimuli applied between 15 different locations, covering an area of 24 cm^2^ [50].

Recent research by Hu et al. [54] focused on the development of a magnetic skin incorporating a 4 × 4 Hall sensor array. The proposed sensor leverages convolutional neural network algorithms to enable precise location perception, achieving an average localization error of 1.2 mm across a 48,400 mm^2^ surface. Additionally, it demonstrates the capability to recognize large-scale shapes and movements across its sensing area.

Previous studies have demonstrated the viability and promise of a multilayered tactile sensing device that integrates magnetic sensors beneath magnetorheological elastomeric layers, effectively mimicking cutaneous mechanoreceptors in human skin. However, this review highlights that the application of magnetoresistive (MR) sensors in such devices remains an understudied area.

Therefore, the primary objective of this research was to develop a magnetoresistive tactile sensor incorporating magnetorheological elastomeric artificial skin. In this system, MR sensors detect magnetic field variations caused by elastomer deformations and translate them into force responses.

To deepen the understanding of the underlying magnetic field changes during material deformation, extensive simulations were conducted. These insights guided the design and testing of a prototype device. Machine learning was used to characterize the magnitude and location of normal and tangential forces. As previously noted, this approach sets itself apart from existing devices by integrating a continuously distributed magnetic field source with TMR sensors, resulting in greater sensitivity when measuring magnetic fields compared to Hall sensors [55]. Beyond introducing a novel technology to tactile sensing and validating its viability for such applications, our work aimed to refine force and spatial predictions, achieving higher accuracy in these measurements.

## 2. Materials and Methods

### 2.1. Elastomer Simulation Model

To understand the mechanical and magnetic behaviors of a magnetorheological elastomer (MRE) piece, a two-stage simulation was developed using COMSOL Multiphysics, focusing on its mechanical deformation and its influence in magnetic field shifts. The simulation system consisted in two distinct stages. The first stage simulated the mechanical deformation of an elastomeric material when subjected to the force applied by an indenter. This simulation aids in the understanding of how the surface of the elastomer displaces for various loads. The second component of the simulation focuses on the magnetic field shifts that originate from these deformations. It explores how changes in the material’s geometry influence the magnetic field distribution.

This section provides a detailed explanation of the simulation model development, including the system’s geometry, its mechanical and magnetic properties, and the meshing parameters.

#### 2.1.1. Geometry

The deformation simulation geometry was designed to match an existing experimental setup, consisting of a cylindrical indenter with a rounded tip and an elastomer membrane. Figure 1 shows this system, along with the respective dimensions and adopted coordinate system. A 2D axisymmetric model was used to reduce the computation time. This means that when the geometry is revolved around the symmetry axis, a circular membrane is produced, in contrast to the square-shaped configuration used in the physical device. This was a necessary compromise that came with no detriment to the results of this simulation.

Once the mechanical simulation had been computed, where the elastomer was deformed by the indenter, the resulting geometry could be exported to magnetic simulation. Some changes were made to enable the calculation of the magnetic field produced by the MRE sample, including the added representation of the air environment, providing the relative permeability of the environment, $\mu_{0}$.

#### 2.1.2. Mechanical and Magnetic Parameters

The mechanical behavior of the elastomer was modeled using the Neo-Hookean framework, with a Young’s modulus of E=125 kPa [56]. The indenter was treated as a rigid body, while constraints were applied to simulate the elastomer resting on a fixed surface. Incremental force was applied in steps of 0.5 N, ranging from 0 N to 5 N.

The magnetic behavior was modeled as a uniformly magnetized volume, with a remanent flux density norm, $\left|\right|B_{r}\left|\right|=$ 0.2434 T, obtained experimentally in Section 2.2.6.

#### 2.1.3. Meshing

The COMSOL simulations resorted to the Finite Element Method (FEM). For all simulations, triangular FEM elements were used. A higher-density mesh was used near the indenter and elastomer interface to ensure a localized increase in spatial resolution where the deformations are more drastic. This balance between mesh size and computation time enhanced the precision of the simulation. The maximum and minimum sizes of the FEM domains are presented in Table 1. Overall, the same meshing strategies were applied in this second simulation stage. However, the mesh was further refined at the air–elastomer interface on the top side, where the field line density is higher and the geometry is more complex due to localized deformation. The element size parameters are detailed in Table 1.

### 2.2. Design, Fabrication, and Characterization of the Device

The development of the envisioned tactile sensor required extensive engineering processes involving magnetoresistive sensors and magnetorheological elastomers. This section outlines the various techniques employed in the planning, fabrication, and characterization of the device. The process includes the design and assembly of the Printed Circuit Board (PCB), sensor chip characterization, integration and encasing, MRE fabrication and analysis, and, ultimately, testing of the final prototype.

#### 2.2.1. Printed Circuit Board

The PCB design was performed using Altium Designer for a 25.5 mm × 25.5 mm square (650.25 mm^2^), 2-layer board. The top side was covered in a 4 × 4 matrix of custom-made double ‘L’ pads, with a spacing of 4.5 mm for rows and columns. This design allows for multiple sensor mounting orientations, which could be useful in future iterations of the device.

Matrix notation was used to identify each sensor; for example, sensor $R_{11}$ represents the first sensor in the first column, positioned in the top left corner. The PCBs’ layout and their details are shown in Figure 2a and Figure 2b, respectively. The separation between adjacent sensors is 4.5 mm.

#### 2.2.2. Magnetic Field Sensors

Sensor dies were fabricated at INESC-MN using tunnel magnetoresistive sensors based on MgO [57,58], with a die size of 3 mm × 0.5 mm. The sensors were based on the following film stacks [59]: [5 Ta/15 Ru ] × 2/5 Ta/5 Ru/17 PtMn/2 CoFe/0.85 Ru/2.6 CoFeB/MgO 1.5/3 CoFeB/0.21 Ta/8 NiFe/8 IrMn/2 Ru/5 Ta/10 Ru (thickness in nanometer), deposited on a 200 mm diameter wafer. The sensor was patterned into 2 × 20 µm pillars connected in series for lower noise and high electrical breakdown resilience [60]. Two orthogonal magnetic field annealing steps were carried out after microfabrication to set the linear response through crossed-anisotropies between the reference and the free layer [61]. The optimization of TMR sensors for industrial and robotics applications has been extensively studied by various authors (e.g., [62,63]), and further reading is encouraged for a deeper understanding.

The sensitive direction was set along their width. Each die had 4 pads, and their nominal resistance was established between the ‘−’ and ‘+’ pads and between the ‘G’ and ‘*▪*’ pads. The dies were individualized in a Disco DAD321 automatic dicing tool, with the sensors characterized upon integration into the PCB with wire bonding. Figure 3 shows the magnetoresistive R(H) curve for one of these sensors, with a sensitivity of 9.427 Ω/Oe in its linear region (between 0 and 120 Oe) and the nominal resistance values at $H_{ext}=0$. The sensor’s maximum hysteresis is 1.5 Oe. The sensor does not reach saturation within fields ranging between −140 and 140 Oe. When measuring in larger field ranges, a maximum TMR ratio of 135% was registered for an applied field of 300 Oe, confirming the high quality of the materials selected for this work.

#### 2.2.3. Wire Bonding

Wire bonding was chosen as the interconnection technique to connect the magnetic sensors onto the PCB. Upon mounting the dies onto the board using Loctite 431 adhesive, the sensors were wedge-bonded using a Kulicke & Soffa 4526 manual bonding system, with a 45 μm aluminum wire. The connection was made under a crossbar matrix architecture, with the ‘+’ terminals of the sensors connected in columns whereas the ‘−’ terminals were connected in rows.

#### 2.2.4. Epoxy Encasing

Wire bonding results in a relatively fragile connection; therefore, an MG Chemicals 832HD black epoxy resin was applied to the sensor chips to ensure the physical integrity of the wires during touch experiments.

A Drifton 2000-D controlled dispenser was used at 0.4 MPa to cover 12 separate bond-wired sensor dies with known but distinct volumes of epoxy. The mechanical robustness of the protection was evaluated upon applied forces to determine the breaking point versus protection thickness. These results are shown and discussed in Section 3.2. As a result of these tests, the device was encapsulated with an epoxy volume of 0.11 mL per chip, providing a balanced compromise between device sturdiness and minimal epoxy use.

#### 2.2.5. Elastomer Fabrication

The artificial skin was fabricated using Ecoflex 00-30, a silicone rubber selected for its flexibility and low Young’s modulus of E=125 kPa [64] when compared to other popular alternatives such as PDMS [6,65]. To create an MRE, Magnequench [66] MQFP-B+ 5 µm NdFeB ferromagnetic particles [67] were blended into Ecoflex at a 60:40 ratio [68,69]. A similar formulation was previously applied in cilia fabrication and successfully evaluated for fine surface characterization [70].

The elastomer mixture underwent a 25 min degassing process in a vacuum desiccator to eliminate air bubbles before being poured into a 3D-printed Polylactic Acid (PLA) mold (see Figure 4a). It was then degassed for an additional 25 min and left to cure at room temperature for 4 h.

The resulting composite is shown inside its mold in Figure 4b.

The orientation of the magnetic domains in the MRE piece can have an impact on the global sensor response [71,72]. Therefore, to test how an anisotropic MRE would perform as a tactile sensor in contrast to an isotropic one, an annealing treatment was performed. The elastomer was heated to 80 °C and then cooled, at room temperature, under a uniform magnetic field of 0.8 T.

#### 2.2.6. Vibrating-Sample Magnetometer Measurements

The magnetic properties of the MRE were measured using a Microsense EZ-9 Vibrating-Sample Magnetometer (VSM). Hysteresis loops were recorded for both isotropic and anisotropic samples, as seen in Figure 5, and the magnetic properties are summarized in Table 2. The hysteresis reduction is the most significant consequence of the magnetic annealing, and it is consistent with similar trends reported by other sources [73]. The other properties remain very similar upon annealing.

#### 2.2.7. Device Characterization

The fully assembled device (see Figure 6) comprises the 4 × 4 TMR sensor matrix encapsulated in epoxy, buried under a 3 mm layer of Ecoflex-based MRE. Initial testing was performed, following the same procedure as the sensor characterization, confirming the proper operation of each sensor. The sensor matrix was measured, and each sensor resistance shows a linear dependence on the magnetic field. These results, shown in Figure 7, confirmed the functionality of the integrated system and set the stage for further experimental data collection under applied forces.

### 2.3. Data Acquisition and Model Training

After the device was fully assembled, data was collected to train machine learning models capable of predicting both the magnitude of forces along three linear axes and their precise locations.

The data collection setup involved a controlled 3D movement stage with an ATI Nano17 force sensor mounted on the Z-axis. This sensor, with a resolution of 12.5 mN, was attached to a PLA indenting tip to apply force to the elastomer surface of the device. Data from both the force sensor and the device were collected simultaneously using a DAQPad-6015 device at 62.5 Hz for the force sensor and a custom acquisition board at 72 Hz for the sensor array. Both devices store data as points in *.csv* files, with 3 columns in the case of the force sensor (one for each of the 3 linear axes—x, y, and z) and 16 in the case of the sensor array (one for each of the individual sensor chips). The sensor array acquisition board, based on the Zero Potential Method (ZPM), leverages AC acquisition for reduced noise and a high-performance carrier suppression system for better accuracy and a wider dynamic range [74]. Figure 8 presents a photograph of the setup, while Figure 9 illustrates the connection scheme for each component within the system.

The data acquisition approach varied depending on the specific objective for which the data was being collected. The distinct methodologies to test the device response and gather data for neural network training are summarized in Figure 10 and are described next.

#### 2.3.1. Magneto-Elastic Response Testing

To characterize the general response of the device, the approach involved measuring how a deformation in a given location affected the magnetic field in the proximity sites. Therefore, a varying localized force was applied, caused by lowering the indenting tip, perpendicularly to the surface of the device and above sensor $R_{33}$. The data was collected from both the force sensor and the magnetic sensor matrix. This procedure was carried out for both the isotropic and anisotropic MRE samples.

#### 2.3.2. 3D Force Characterization

To gather the necessary data to train the neural network (NN) model in 3D force sensing, the indenter was set to apply forces in the *x*, *y*, and *z* directions, maintaining the indenter in contact with the elastomer surface. Figure 10-top illustrates the methodology. The movement in these axes allowed for data collection for both normal and tangential forces to be used in model training. It is important to note that the latter consistently induced shear strain in the elastomer without causing a lateral displacement of the indenter across its surface.

Since the data acquisition rates of the force sensor and the device differed, preprocessing was necessary to synchronize the two measurements. Following standard neural network training procedures, the dataset was divided into training and test sets, consisting of 90% and 10% of the total data, respectively.

Other machine learning models, such as the random forest regressor, were also considered and tested. However, their performance was significantly inferior to that of the Multilayer Perceptron (MLP), which was ultimately selected. Likewise, various train/test split ratios were explored, and a 90/10% split was chosen as it yielded the most accurate predictions.

The MLP model included three dense layers: two with 128 neurons and one with 32, all using Rectified Linear Unit (ReLU) activation and L2 regularization. They were followed by batch normalization and dropout layers to prevent overfitting. The final output layer, consisting of three neurons with linear activation, computes the force magnitude for each dimension.

#### 2.3.3. Spatial Awareness

The data required to train an MLP with spatial awareness was collected using a procedure similar to response testing, as illustrated in Figure 10-bottom. However, this process was repeated for each sensor location, with all 16 locations assigned a unique set of (*x*,*y*) coordinates, originating from the center of the device’s surface.

The MLP model was constructed similarly to the force-sensing variant but with fewer neurons in the dense layers and a higher dropout rate. To accommodate the 2D nature of the data, the output layer consists of only two neurons.

## 3. Results

This section presents the most relevant findings of the study. First, the results of the mechanical and magnetic simulations are examined. The outcomes of the two-stage simulation model are analyzed within the context of experimental procedures, contributing to their validation. The predicted deformation of the elastomeric membrane in the mechanical stage is directly correlated with the magnetic field variations estimated in the magnetic stage.

Next, the results of the structural integrity tests are presented, highlighting their relationship with the chip encapsulation volume. This is followed by an analysis of the device’s direct response, concluding with a discussion on the application of Multilayer Perceptron models for characterizing the stimuli applied to the device.

### 3.1. Simulations

The mechanical simulation aimed to evaluate how a localized transverse force applied to the elastomeric surface can affect the deformation across the device. To better visualize the impact of the deformation throughout the entire surface, four zones were defined, A through D, which are equidistant from the point of contact and coincident with the sensor location, as seen in Figure 11. The resulting plot can be seen in Figure 12. The deformation values for these distances are shown in Table 3.

This analysis serves to infer that, according to the simulation model, there is a clear distinction in the deformation that occurs in points at various distances from the contact point. Even for less drastic deformations caused by 0.5 N and 1 N, the shift in the elastomeric surface is noticeable, translating in a 3.7% and 7.1% relative shift, respectively, for the closest sensors in zone A.

The next stage in the simulation model consisted in modeling the influence of the elastomer magnetic field on the sensor matrix, both before and after a deformation. Only the radial component of the magnetic field, $\vec{H_{r}}$, is relevant to this analysis. The axial component, $\vec{H_{z}}$, is not sensed by the TMR chip, and the tangential component, $\vec{H_{\theta }}$, is comparatively negligible.

In Figure 13, $\vec{H_{r}}$ is plotted along the *r* axis for three elastomer configurations. The field values felt at distances A through D are displayed in Table 4. These results show that the sensors located near the force application point experienced the most significant shifts in the magnetic field. Even low-magnitude deformations produce distinct localized changes of 172.9% and 323.7%, further validating the device for force characterization purposes. Furthermore, the results assure that the sensors will operate within the linear region at all times, as the local maximum magnitude of the magnetic field felt by any chip is 39.50 Oe and this value tends to decrease for harsher deformations.

### 3.2. Impact of the Volume of Epoxy on the Robustness of the Device

As discussed in detail in Section 2.2.4, it was crucial to determine the physical limits of the wire bonding connection when encapsulated with epoxy. After determining the flow rate of the fluid to accurately quantify the volume of epoxy that was dispensed for each chip, they were submitted to progressively stronger pressures until a breaking point was reached, with the setup described in Section 2.3.

The results, in Figure 14, show that, as expected, sensors with more epoxy could withstand higher forces, with maximum forces ranging from 10 N to 150 N, depending on the amount of epoxy applied. These findings informed a more evidence-based determination of the optimal epoxy deposition volume for the prototype. The objective was to achieve a balance between minimizing the amount of epoxy and ensuring adequate structural integrity under direct forces of up to 20 N. As discussed in Section 2.2.4, this analysis led to the adoption of an encapsulation volume of 0.11 mL per sensor chip.

### 3.3. Response of the Device to Applied Force

This section focuses on exploring the main objective of this project, analyzing the response of the device when a force is applied to its surface. The direct response of the device to forces of varying magnitudes is presented and examined, followed by a detailed evaluation of its integration with the neural network models.

#### 3.3.1. Characterization

The response of the device to applied forces was tested to measure how deformations in the elastomeric layer affected nearby sensors. A force, localized to $R_{33}$, of up to 1.04 N was applied, as seen in Figure 15a. The data, presented in Figure 16 and Figure 17, reveals a distinctive response from sensors directly adjacent to the contact point, since these display the largest resistance variance. In particular, it should be emphasized that $R_{32}$ and $R_{34}$ exhibit the most drastic deviations from their nominal resistance. This result is entirely corroborated by previous considerations made evident by the simulation model, which predicts that the largest shift in the magnetic field will occur for its radial component (see Figure 13).

Given the orientation of the sensors on the board and their sensitivity axis, which is perpendicular to their longitudinal direction, these sensors are expected to exhibit the most pronounced resistance changes. Since the largest changes in the magnetic field happen radially from the point of contact, it follows that sensors that are mounted perpendicularly to this direction experience a larger shift in resistance when compared to the ones mounted longitudinally.

The same procedure was then carried out for the annealed Ecoflex sample. As shown in Figure 18, the response was similar, with $R_{32}$ and $R_{34}$ once again exhibiting the largest shifts from their resting positions, though slightly less pronounced than before. Given these results—and considering that the isotropic sample successfully generated an adequate response—a decision was made to exclusively utilize this sample for the subsequent objectives. This choice is further supported by previous VSM analyses and prior research on the topic ([72,73]).

#### 3.3.2. 3D Force Characterization

The accuracy of the force-sensitive MLP, described in Section 2.3.2, was evaluated using the 10% of the data that did not enter the training set. In this dataset, shown in Figure 19, the model exhibited a coefficient of determination R^2^ = 0.9624, MSE of 0.0187 N^2^, and a mean absolute error (MAE) of 0.0843 N between the predictions and the actual values. Additionally, the model was applied to other independent sets, exhibiting a minimum MAE of 0.07 N and a maximum of 0.17 N. Figure 20 shows one of these analyses. Overall, the results indicate that the device can accurately quantify forces applied in three dimensions, making it reliable for tactile sensing applications.

#### 3.3.3. Spatial Awareness

The test set was used once again to make inferences about the accuracy of the model. As one can see in Figure 21a, it was fairly accurate in the predictions of the test set, yielding an MSE of 0.1017 mm^2^, $R^{2}$ = 0.9958, and an MAE of 0.2552 mm. When predictions are mapped to the nearest of the 16 tested locations, the accuracy reaches 100%; i.e., the device consistently distinguishes between 16 distinct points across an area of 25.5 mm × 25 mm. The maximum and minimum registered euclidean distances between the real values and the predictions were 1.0692 and 0.0018 mm, respectively. The ability of the model to predict locations outside its training scope was tested as well. For that purpose, the indenting tip exerted force on the center of the elastomeric membrane, as seen in Figure 15b. The resulting prediction, seen in Figure 21b, reveals a minimum euclidean distance to the actual force application point of 0.8894 mm and a maximum of 1.1609 mm, and an MAE = 0.6303 mm. Overall, it performed well on predicting locations of an applied stimulus and was also capable of extrapolating for unrepresented ones.

## 4. Discussion

In this study, a tactile sensor was designed with a biomimetic architecture where a magnetorheological elastomer acts as an artificial skin, while deformations are detected by a 4 × 4 TMR sensor matrix. Simulations guided the device’s design, modeling how an applied force deforms the 3 mm thick skin and impacts the surrounding magnetic field.

To ensure the structural integrity of the wire-bonded sensors encapsulated in epoxy, comprehensive mechanical tests were conducted. Multiple sensor dies, identical to those that were used in the device, were covered with varying volumes of resin. They were subjected to progressively stronger localized pressures until they reached their rupture point. The tests determined that an encapsulation with 0.12 mL of epoxy per die could withstand forces greater than 100 N, making it suitable for high-stress applications. The sensors were then encased based on these findings.

A 3D stage with force sensors was employed to collect data from the device for the training of a neural network model. The amplitude sensing model was able to consistently quantify both normal and tangential forces, with an MAE of 0.08 N and an MSE of 0.02 N^2^. The spatially sensitive model was trained using 171,008 points taken at the 16 different locations. The model was able to successfully distinguish the location of an applied pressure between these points, offering predictions with an MAE of 0.26 mm and an MSE of 0.10 mm^2^, for a sensitive area of 25.5 mm × 25.5 mm with sensors 4.5 mm apart. It was also able to extrapolate to areas outside its training scope with an error of 0.63 mm.

The scores obtained are generally more promising than the previous studies presented in Section 1, with devices based on Hall-effect magnetic sensors. In comparison to ReSkin [45], which reported a mean squared error of 0.142 N^2^ for force estimation and 0.514 mm^2^ for spatial awareness, the proposed device achieved better results in both aspects. It also surpassed the force sensing accuracy of the system developed by Hellebrekers et al. [49], which reported a minimum mean error of 0.25 N, and achieved results comparable to those of Fang et al. [50], whose device exhibited a force MSE of 0.04 N^2^. Regarding the device’s ability to predict the location of applied stimuli, comparisons with prior studies are less straightforward due to differences in sensing area and taxel density. Nonetheless, the proposed device underperformed in comparison to the system developed by [49], which was able to distinguish between 25 discrete points within a compact 15 mm^2^ area with an accuracy exceeding 98%. However, it outperformed the device by Fang et al., which is reported to distinguish between 15 contact points across a substantially larger 24 cm^2^ area. When compared to the uSkin sensor [48], a commercially available and well-established product, it was not possible to surpass its performance metrics. However, considering the inherently higher sensitivity of TMR technology relative to Hall-effect sensors, there is a strong rationale to believe that future developments may close this gap and potentially exceed current commercial standards.

These results highlight the advantages of adopting TMR technology for tactile sensing and, thus, demonstrate the potential of a sturdy biomimetic tactile sensor built from simple uniaxially sensitive TMR chips that accurately detect both the magnitude and location of forces applied in three dimensions.

## Figures and Tables

**Figure 1 sensors-25-03724-f001:**
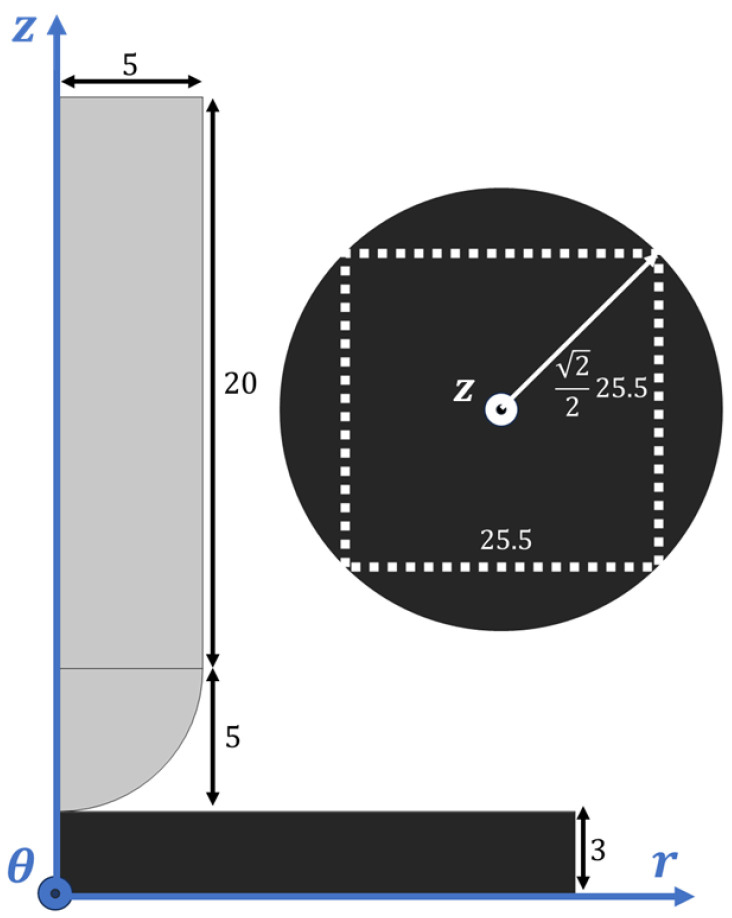
A 2D representation of the geometry utilized during the mechanical deformation simulation. Two solid bodies comprise the model: the indenting tip, in grey, and the elastomer membrane, in black. A conceptual top view of the membrane is also represented, where one can visualize the resulting circular surface alongside the outline of the real elastomer pieces. The dimensions of each object are expressed in millimeters.

**Figure 2 sensors-25-03724-f002:**
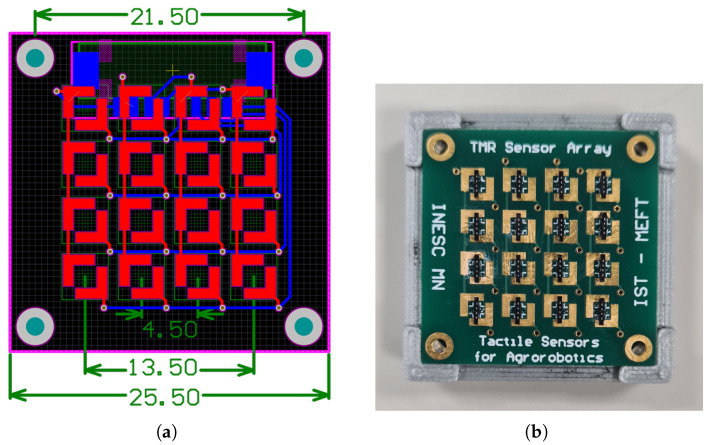
(**a**) Layout view of the PCB with the ‘L’ pads for the TMR sensors. The relevant dimensions are presented in millimeters. (**b**) Picture of the PCB with the TMR sensors.

**Figure 3 sensors-25-03724-f003:**
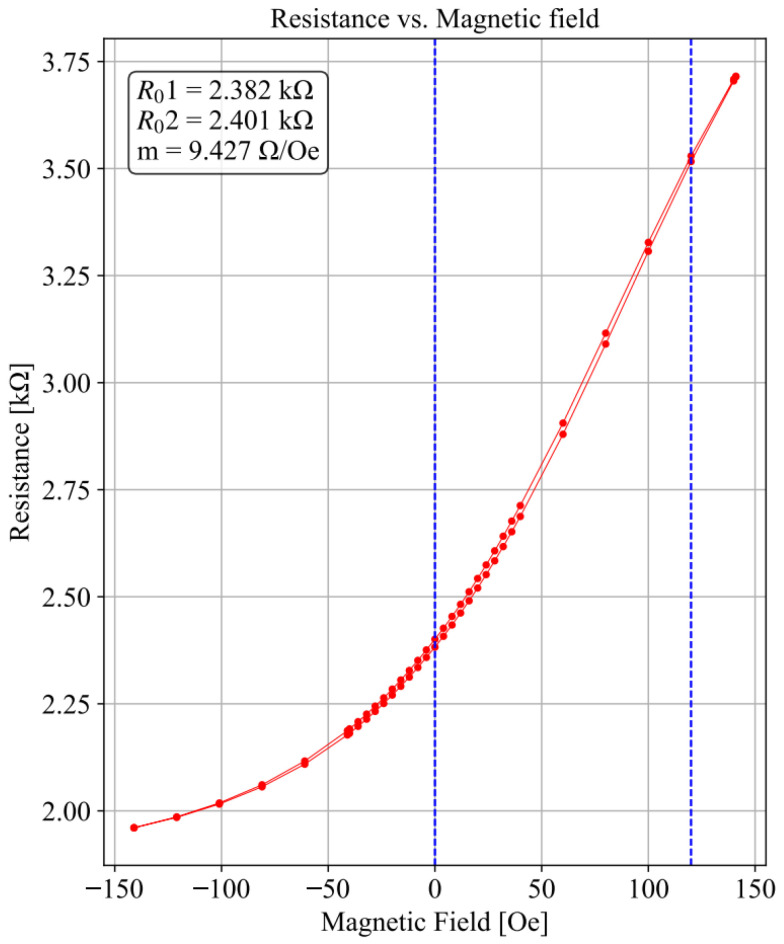
Example of sensor resistance measured across the + and − terminals as a function of an external magnetic field. The individual magnetic field strength values that were employed are shown as red dots (the line connecting them is merely illustrative). The nominal resistance values at a zero field, $R_{0}1$, and $R_{0}2$ are obtained, as well as the sensor’s sensitivity *m*, calculated for the linear region [0, 120] Oe (highlighted in blue). The maximum hysteresis is 1.5 Oe.

**Figure 4 sensors-25-03724-f004:**
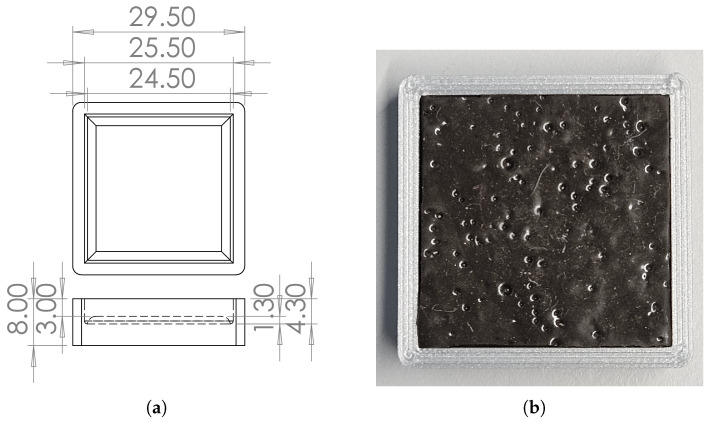
(**a**) Top–down and lateral views of the mold used for the elastomer. Dimensions in millimeter. (**b**) Photo of the elastomer piece inside the mold.

**Figure 5 sensors-25-03724-f005:**
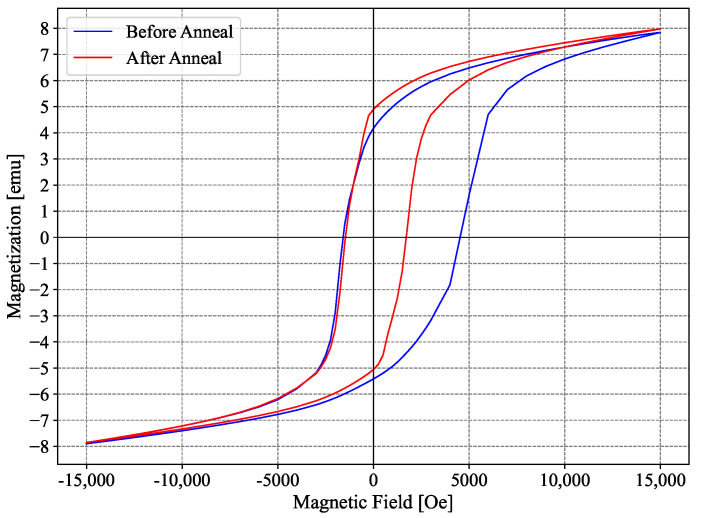
Magnetic moment of the Ecoflex samples measured by VSM.

**Figure 6 sensors-25-03724-f006:**
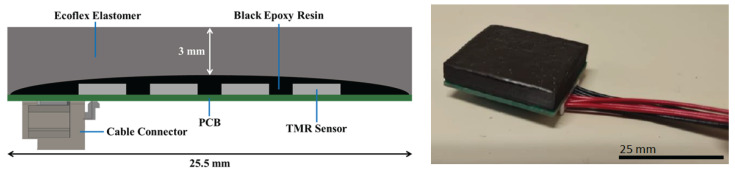
**On the left:** Schematic cross-section of the tactile sensor module, including 16 magnetoresistive sensor chips encapsulated with epoxy and buried underneath a 3 mm thick Ecoflex skin. **On the right:** Photograph of the integrated device.

**Figure 7 sensors-25-03724-f007:**
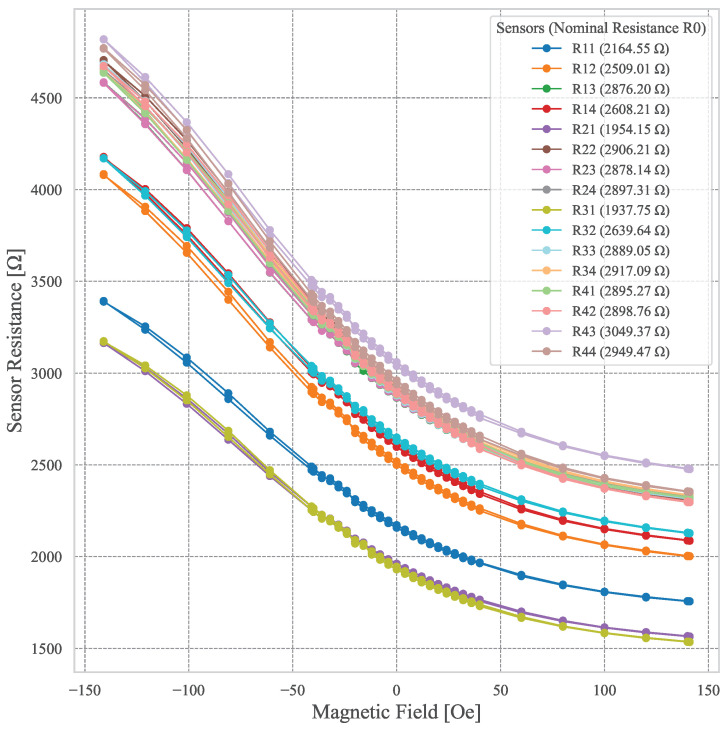
Plot of the resistance across every sensor when an external magnetic field was applied, ranging from −140 to 140 Oe. The nominal resistance values for H = 0 are also shown.

**Figure 8 sensors-25-03724-f008:**
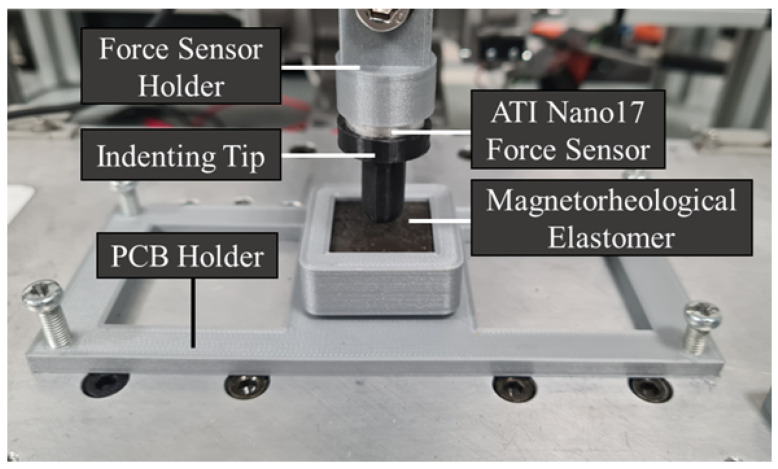
A close-up of the device, indenting probe, and force sensor.

**Figure 9 sensors-25-03724-f009:**
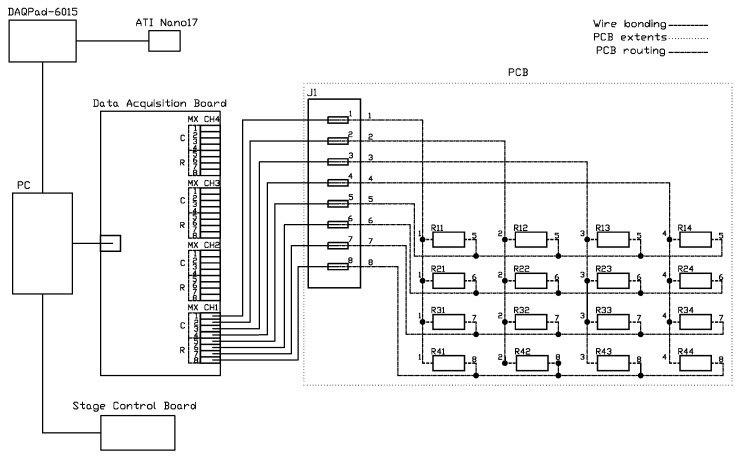
Schematics of the data acquisition setup, indicating the connection type.

**Figure 10 sensors-25-03724-f010:**
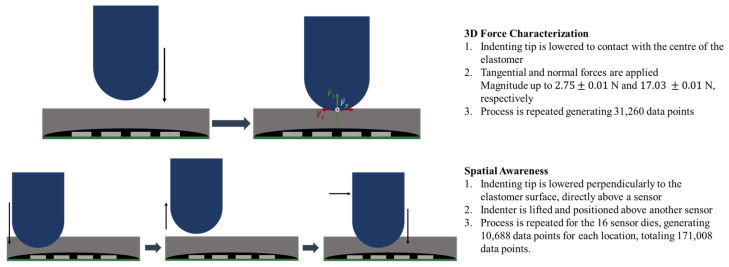
Methods of the data collection for the 3D force characterization (**top**) and the spatial awareness (**bottom**).

**Figure 11 sensors-25-03724-f011:**
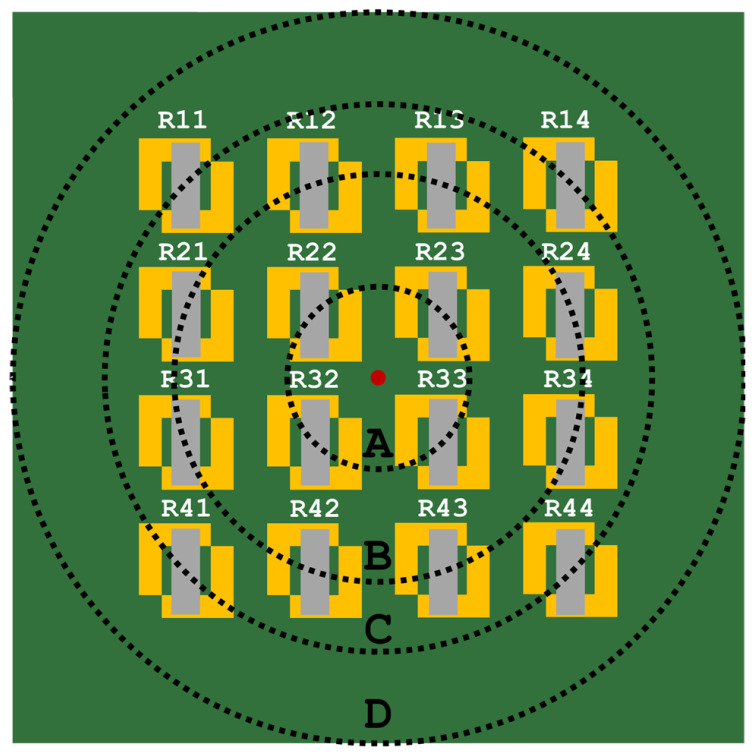
Diagram of the sensors mounted onto the PCB, along with the representation of the contact point and the circumferences A, B, C, and D.

**Figure 12 sensors-25-03724-f012:**
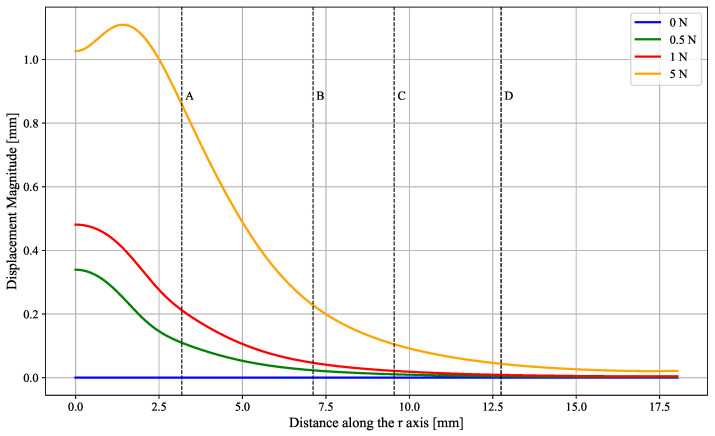
Plot of the displacement magnitude, in mm, of the elastomeric surface across its surface, with respect to its resting position when different forces are applied by the indenting tip.

**Figure 13 sensors-25-03724-f013:**
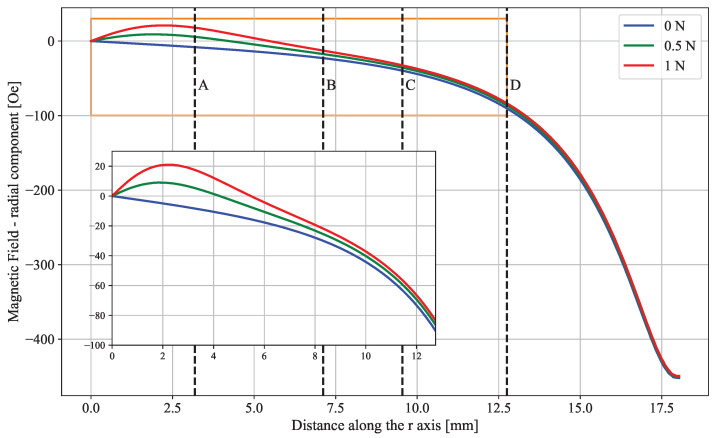
Plot of the radial component of the magnetic field for z=−1.3 mm when different forces are applied by the indenting tip. A zoomed-in section of the plot is presented, ranging from the center of the device to the extents of zone D.

**Figure 14 sensors-25-03724-f014:**
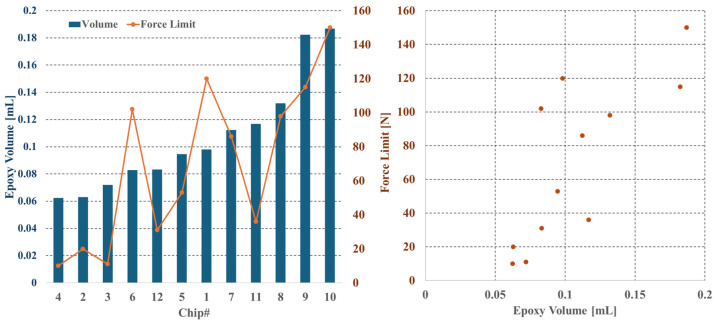
**On the left:** Plot of the volume of epoxy deposited onto each die (displayed as bars) and the maximum force they could withstand before breaking (shown as points). **On the right:** Plot of the maximum force withstood by each sensor and the respective volume of epoxy that was used.

**Figure 15 sensors-25-03724-f015:**
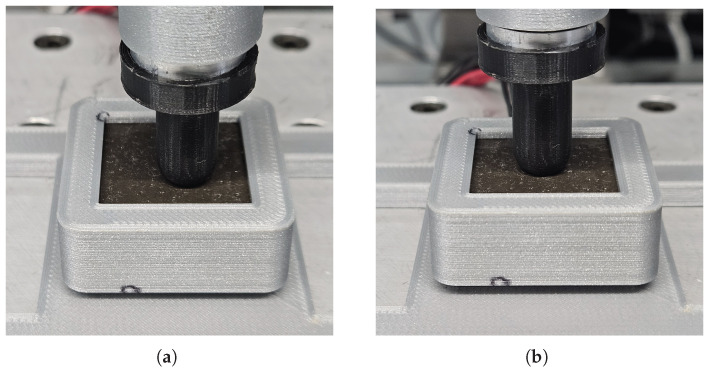
(**a**) Photograph of the indenting tip exerting a perpendicular force on the elastomer membrane above sensor $R_{33}$. (**b**) Photograph of the indenting tip exerting a perpendicular force on the center of the elastomer membrane.

**Figure 16 sensors-25-03724-f016:**
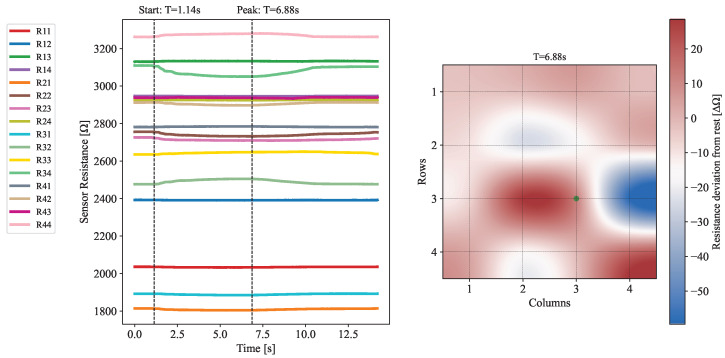
**On the left:** Plot of the evolution of the electrical resistance across each one of the 16 sensors through time when subject to an applied force. Timestamps for the start of the deformation and the time of its peak in magnitude are also represented. **On the right:** Heatmap showing the deviation in the electrical resistance of each sensor with respect to a resting position. The force application point is shown in green. The data shown relates to the time instant when the deformation was at its peak.

**Figure 17 sensors-25-03724-f017:**
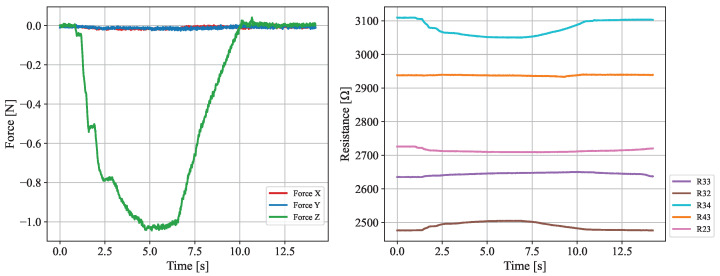
**On the left:** 3-axis force applied on the device surface as a function of time, in seconds. **On the right:** Resistance measured for 5 neighboring sensors.

**Figure 18 sensors-25-03724-f018:**
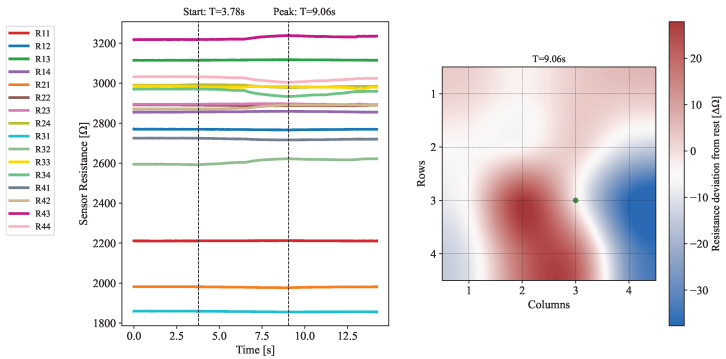
**On the left:** Evolution of the individual sensor resistance upon force application on the 16 sensors under the annealed MRE elastomer piece. The onset for the deformation and the moment with maximum deformation are also represented. **On the right:** Heatmap showing the difference between sensor resistance and the resting position. The force application point is shown in green. The data refers to the moment of maximum deformation.

**Figure 19 sensors-25-03724-f019:**
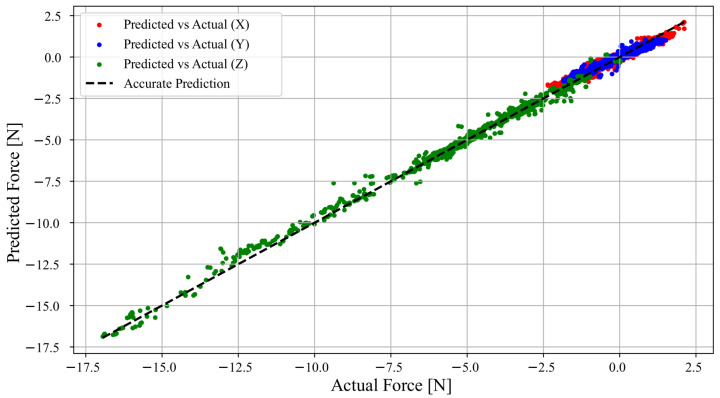
Plot of 3126 points of force that were obtained from the model vs. the actual force values from the test set, in Newton. The force along the X, Y, and Z axis is plotted along with the line that represents a perfect match between the predicted and real values.

**Figure 20 sensors-25-03724-f020:**
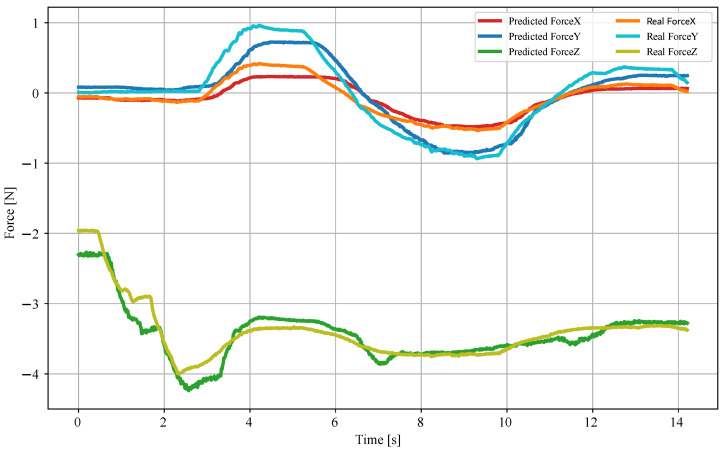
Plot of the force predictions along with the real force values from a variable load through time, for each axis X, Y, and Z, in Newton. The metrics for this particular set are MAE_*X*_ = 0.07 N, MAE_*Y*_ = 0.12 N, and MAE_*Z*_ = 0.12 N.

**Figure 21 sensors-25-03724-f021:**
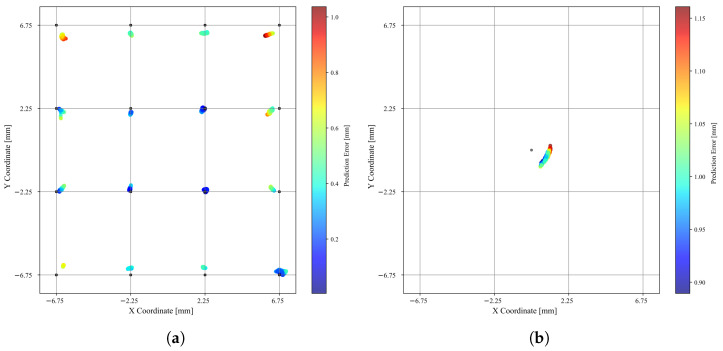
MLP performance on locating applied forces. (**a**) A 2D plot of the 17101 points, as predicted by the model, color-coded by their euclidean distance to the actual coordinate, shown in black. (**b**) Coordinate predictions outputted by the MLP when force was applied on coordinate (0,0).

**Table 1 sensors-25-03724-t001:** Relevant mesh parameters for both the mechanical and magnetic simulations.

Parameter	Mechanical Max [mm]	Mechanical Min [mm]	Magnetic Max [mm]	Magnetic Min [mm]
Global	0.3	6 × $10^{-3}$	2	4 × $10^{-3}$
Indenter Surface	0.05	1.3 × $10^{-3}$	N/A	N/A
Elastomer Surface	0.02	5.6 × $10^{-4}$	N/A	N/A
Air–Elastomer Interface	N/A	N/A	0.05	4 × $10^{-3}$

‘N/A’ indicates that the region was not meshed in that simulation domain.

**Table 2 sensors-25-03724-t002:** Average magnetic properties of the sample before and after annealing.

Parameter	Pre-Annealing	Post-Annealing
Coercive Field—$H_{c}$ [Oe]	3055.874	1588.129
Saturation Magnetization—$M_{s}$ [emu]	7.870	7.914
Saturation Flux Density—$M_{s}$ [T]	0.3850	0.3871
Remanent Magnetization—$M_{r}$ [emu]	4.794	4.976
Remanent Flux Density—$M_{r}$ [T]	0.2345	0.2434

**Table 3 sensors-25-03724-t003:** Displacement magnitude, in millimeters, across four zones equidistant from the center of the elastomer surface. The percentual displacement, relative to the thickness of the elastomer membrane, is also shown.

	Absolute [mm]				Relative [%]			
	**A**	**B**	**C**	**D**	**A**	**B**	**C**	**D**
0 N	0	0	0	0	0.0	0.0	0.0	0.0
0.5 N	0.1100	0.0230	0.0106	0.0043	3.7	0.8	0.4	0.1
1 N	0.2126	0.0463	0.0213	0.0087	7.1	1.5	0.7	0.3
5 N	0.8605	0.2265	0.1057	0.0437	28.7	7.6	3.5	1.5

**Table 4 sensors-25-03724-t004:** Radial component of the magnetic field, in Oersted, across four zones equidistant from the center of the elastomer surface. The percentage change in the magnetic field is also shown relative to the magnitude of the magnetic field when the elastomer is at rest.

	Absolute [Oe]				Relative [%]			
	**A**	**B**	**C**	**D**	**A**	**B**	**C**	**D**
0 N	−7.9891	22.8974	39.4987	90.3519	N/A	N/A	N/A	N/A
0.5 N	5.8241	17.4475	35.4187	85.8504	172.9	23.8	10.3	5.0
1 N	17.8722	12.9115	32.0997	81.9019	323.7	43.6	18.7	9.4

## Data Availability

Data that support the findings of this study is available on request from the corresponding author.

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
