# Peer review of "Skin-Inspired Magnetoresistive Tactile Sensor for Force Characterization in Distributed Areas"

_sensors, 2025, doi:10.3390/s25123724_

Round 1

Reviewer 1 Report

Comments and Suggestions for Authors

This study aimed to develop a biomimetic, skin-inspired tactile sensor device capable of sensing applied force, characterizing it in three dimensions, and determining the point of application. This manuscript demonstrates high overall quality and presents both innovative ideas and practical potential. Nevertheless, there are a few minor issues that should be addressed before publication.

1、Before presenting the main objectives of this study in the final paragraph of the “Introduction”, it is recommended that the authors add a brief summary of the preceding content. Currently, the transition from the discussion of recent related works to the statement of this study’s goals appears somewhat abrupt and lacks smooth logical flow. Including a concise summary sentence that reviews the research background and current challenges discussed earlier—and then naturally leads into the necessity and innovation of the present study—would greatly enhance the coherence and readability of the Introduction.    

2、In the introduction to Section 3 “Results,” the authors merely state that “This section displays the most relevant results obtained from this study.” This description is overly brief. It is recommended that the authors expand this section by clearly indicating which specific tests or simulation stages each subsection corresponds to. The introduction of a section should serve a summarizing and guiding role, helping readers understand the structure and logical flow of the subsequent content. An insufficient introduction can hinder readers’ comprehension of the research process and article organization. Moreover, it is suggested that the authors enhance the introductory content in other sections as well—such as Sections 2.1, 2.2, and 3.3—by adding brief background information or clarifying the purpose of the subsection, to improve coherence and overall readability.    

3、The analysis in Section 3.2 appears rather thin, as it mainly presents the results without sufficient discussion or interpretation. It is recommended that the authors enrich this section with more analytical commentary.    

4、The authors do not appear to clearly explain how the data was collected or what specific types of data were acquired during the data collection phase. It is recommended that this section be improved by providing more detailed information on the data acquisition process and the nature of the collected data.    

5、In Section 3.2.2, the authors mention that 90% of the data is used for training and 10% for testing, but do not explain the specific reasons for this division. Was this choice made because of the large amount of data collected? It is also unclear whether the data were randomly assigned and whether cross-validation was performed. This lack of information may raise concerns about the reliability of the model assessment. In addition, the authors chose to use the MLP model for training and prediction, but did not provide a rationale for choosing this model over other possible machine learning methods. It is recommended that the authors clarify the rationale behind the data segmentation strategy and model selection.    

6、The authors state: "Once the device was fully assembled, data was collected to train machine learning models for predicting the magnitude, direction, and location of applied forces." However, the paper only appears to present predictions for force magnitude and location, with no discussion or results related to the prediction of force direction. This discrepancy should be addressed for consistency.

7、In Table 1, the author uses "∙" to represent "×", which does not conform to the standard format of scientific notation. Please correct this. In addition, there seems to be a confusion between horizontal lines and minus signs in all tables and figures, such as Table 1 and Figure 3. Please carefully check and revise the rest accordingly.

8、The fonts used in the figures do not appear to be entirely in Times New Roman and seem rather inconsistent. Please check carefully.

9、Although the use of the TMR sensor is proposed, the manuscript lacks a systematic comparison with existing devices such as ReSkin and uSkin. Please consider adding a comparative analysis to better highlight the advantages of the TMR sensor.

10、Some sections, such as 3.3.1, contain informal or conversational language. It is recommended to refine the wording for improved academic rigor and logical clarity, especially in the data description sections, which could be revised to more standardized scientific expressions.

Comments on the Quality of English Language

The English expressions may be suitably refined for clarity and improved academic tone.

Author Response

Comments 1: Before presenting the main objectives of this study in the final paragraph of the “Introduction”, it is recommended that the authors add a brief summary of the preceding content. Currently, the transition from the discussion of recent related works to the statement of this study’s goals appears somewhat abrupt and lacks smooth logical flow. Including a concise summary sentence that reviews the research background and current challenges discussed earlier—and then naturally leads into the necessity and innovation of the present study—would greatly enhance the coherence and readability of the Introduction.    

Response 1: We would like to thank the reviewer for the helpful and thorough feedback. We agree and have accordingly revised the last paragraph in section 1, which now provides a smoother transition from one topic to the other.

Comments 2: In the introduction to Section 3 “Results,” the authors merely state that “This section displays the most relevant results obtained from this study.” This description is overly brief. It is recommended that the authors expand this section by clearly indicating which specific tests or simulation stages each subsection corresponds to. The introduction of a section should serve a summarizing and guiding role, helping readers understand the structure and logical flow of the subsequent content. An insufficient introduction can hinder readers’ comprehension of the research process and article organization. Moreover, it is suggested that the authors enhance the introductory content in other sections as well—such as Sections 2.1, 2.2, and 3.3—by adding brief background information or clarifying the purpose of the subsection, to improve coherence and overall readability.

Response 2: We agree and have accordingly provided a more detailed introduction for sections 2.1, 2.2, 2.3 and 3.3, where the contents of said sections are summarized and contextualized.

Comments 3: The analysis in Section 3.2 appears rather thin, as it mainly presents the results without sufficient discussion or interpretation. It is recommended that the authors enrich this section with more analytical commentary.

Response 3: We thank the reviewer for this observation. Section 3.2 has been revised to include a more detailed analysis of the relationship between epoxy volume and mechanical robustness, supported by quantitative data and interpretation.

Comments 4: The authors do not appear to clearly explain how the data was collected or what specific types of data were acquired during the data collection phase. It is recommended that this section be improved by providing more detailed information on the data acquisition process and the nature of the collected data.

Response 4: Section 2.3 has been revised to more clearly state the nature of the data files that were collected. Furthermore, to better illustrate the data collection process, a photograph of the experimental setup was added (Figure 8) in addition to a schematic representation of the process (Figure 10) and minor additions were made to the content in sections 2.3 and 2.3.1.

Comments 5: In Section 3.2.2, the authors mention that 90% of the data is used for training and 10% for testing, but do not explain the specific reasons for this division. Was this choice made because of the large amount of data collected? It is also unclear whether the data were randomly assigned and whether cross-validation was performed. This lack of information may raise concerns about the reliability of the model assessment. In addition, the authors chose to use the MLP model for training and prediction, but did not provide a rationale for choosing this model over other possible machine learning methods. It is recommended that the authors clarify the rationale behind the data segmentation strategy and model selection.  

Response 5: We acknowledge that the absence of cross-validation may limit the robustness of the model assessment. However, the primary objective of this study was not to conduct a systematic comparison of different machine learning models or configurations. Nevertheless, various models and configurations were explored during preliminary experimentation, and the MLP was ultimately selected since it provided the best performance in our specific context. To address these considerations, a paragraph discussing the rationale behind the model choice and data split strategy has been added to Section 2.3.1. We also recognize the value of a more rigorous, quantitative comparison and plan to include a comprehensive cross-validation analysis comparing distinct machine learning models in future work.

Comments 6: The authors state: "Once the device was fully assembled, data was collected to train machine learning models for predicting the magnitude, direction, and location of applied forces." However, the paper only appears to present predictions for force magnitude and location, with no discussion or results related to the prediction of force direction. This discrepancy should be addressed for consistency.

Response 6: We appreciate your observation, however, we consider that the ability of the model to accurately predict the magnitude of each of the 3 components (x, y, and z) of applied forces constitutes a prediction of the direction of said force. Furthermore, the model is able to distinguish not only the magnitude of each individual component, but also its sign (positive or negative) as seen in Figure 19. However, we have changed the phrasing to "predicting the magnitude in three linear axes, and the location of applied forces" so that it provides a clearer explanation.

Comments 7: In Table 1, the author uses "∙" to represent "×", which does not conform to the standard format of scientific notation. Please correct this. In addition, there seems to be a confusion between horizontal lines and minus signs in all tables and figures, such as Table 1 and Figure 3. Please carefully check and revise the rest accordingly.

Response 7: Thank you for pointing out these inconsistencies. We have now corrected them to conform to the required format.

Comments 8: The fonts used in the figures do not appear to be entirely in Times New Roman and seem rather inconsistent. Please check carefully.

Response 8: We appreciate the feedback and have changed the fonts to improve the visual consistency of the article.

Comments 9: Although the use of the TMR sensor is proposed, the manuscript lacks a systematic comparison with existing devices such as ReSkin and uSkin. Please consider adding a comparative analysis to better highlight the advantages of the TMR sensor.

Response 9: We agree and have added a comparative analysis in the discussion section, highlighting the performance of the proposed device against ReSkin, uSkin, and others, and noting the future potential of TMR sensors.

Comments 10: Some sections, such as 3.3.1, contain informal or conversational language. It is recommended to refine the wording for improved academic rigor and logical clarity, especially in the data description sections, which could be revised to more standardized scientific expressions.

Response 10: We thank the reviewer for the comment. Section 3.3.1, in particular, underwent substantial revisions to eliminate informal phrasing and improve clarity. Additionally, the entire manuscript was reviewed to ensure consistency with academic writing standards throughout.

Reviewer 2 Report

Comments and Suggestions for Authors

The paper provides a fairly detailed description of the creation and calibration of a matrix Magnetoresistive tactile sensor. All the methods proposed by the authors are adequate and understandable.

Remarks:

  1. Figure 1. It is unclear what is shown in the picture. Do I understand correctly that the figure shows a ball indenter with a radius of 5 mm, and its contact with an elastomer with a thickness of 3 mm. If you are using a two-dimensional axisymmetric model (cylindrical coordinate system), then why do the coordinates r, 0, z have a vector icon? Clearly describe in the caption where the indenter is and where the elastomer is of the sensor system (maybe they can be made in different colors).
  2. Paragraph 2.2.2. Indicate in the magnetic field where the TMR reaches 135%

Author Response

Comments 1: Figure 1. It is unclear what is shown in the picture. Do I understand correctly that the figure shows a ball indenter with a radius of 5 mm, and its contact with an elastomer with a thickness of 3 mm. If you are using a two-dimensional axisymmetric model (cylindrical coordinate system), then why do the coordinates r, 0, z have a vector icon? Clearly describe in the caption where the indenter is and where the elastomer is of the sensor system (maybe they can be made in different colors).

Response 1: Thank you for your helpful comments. Yes, your understanding is correct. To more clearly describe the simulation model, the caption has been revised, as well as the introduction to section 2.1 and the contents in section 2.1.1. The vector icon has also been removed.

Comments 2: Paragraph 2.2.2. Indicate in the magnetic field where the TMR reaches 135%

Response 2: We have included the magnitude of the applied magnetic field for which a TMR of 135% was registered. It now states in section 2.2.2. that "a maximum TMR ratio of 135 % was registered for an applied field of 300 Oe".

Reviewer 3 Report

Comments and Suggestions for Authors

This paper presents a study on the development of a magnetism-based tactile sensor through the design of a TMR sensor array. It explains that high sensitivity was achieved using a neural network (NN) model, and high spatial recognition capability was demonstrated using a multilayer perceptron (MLP) model. However, additional empirical evidence and in-depth discussion are necessary to support the authors’ claims. Furthermore, the paper lacks sufficient explanation of the experimental methods, and clearer illustrations and descriptions are needed.

Overall, revisions are required before this work can be considered for publication.

The authors are expected to address the following comments for further illustrations and clarifications:

1. This study aims to develop a biomimetic, skin-inspired tactile sensing device. However, has this aspect been sufficiently addressed in the paper? The authors should clarify in what ways the sensor is biomimetic.

2. The description of the TMR sensor throughout the paper is insufficient. While the authors claim that higher performance was achieved compared to Hall sensing methods using a TMR sensor, they should also provide a detailed explanation of the TMR sensor itself. Moreover, it would be beneficial to compare it with other TMR-based sensors as well as similar technologies such as AMR and GMR.

3. Is Figure 1 sufficient to supplement the simulation explanation? It would be better to clearly express what kind of simulation was intended through this figure.

4. It would be helpful to include setup photos and results images from the experiments. It is difficult to understand the experimental procedures through diagrams and charts alone.

5. Comparative analysis between actual experiment photos and data results would strengthen the paper. For instance, comparing images of force being applied to the sensor with how that force was recognized would help support the findings.

Author Response

Comments 1: This study aims to develop a biomimetic, skin-inspired tactile sensing device. However, has this aspect been sufficiently addressed in the paper? The authors should clarify in what ways the sensor is biomimetic.

Response 1: Thank you for pointing that out. To address it, the final paragraph of section 1 has been expanded to include a comparison with biological skin: "a multilayered tactile sensing device that integrates magnetic sensors beneath magnetorheological elastomeric layers, mimicking cutaneous mechanoreceptors in human skin".

Comments 2: The description of the TMR sensor throughout the paper is insufficient. While the authors claim that higher performance was achieved compared to Hall sensing methods using a TMR sensor, they should also provide a detailed explanation of the TMR sensor itself. Moreover, it would be beneficial to compare it with other TMR-based sensors as well as similar technologies such as AMR and GMR.

Response 2: We agree that the sensors needed further clarification and have added a more detailed and technical analysis of the TMR sensor that was used in section 2.2.2.

Comments 3: Is Figure 1 sufficient to supplement the simulation explanation? It would be better to clearly express what kind of simulation was intended through this figure.

Response 3: We agree that a more detailed explanation of the simulation was necessary. To address this, the introduction of Section 2.1 has been expanded, and additional clarifications have been incorporated into Section 2.1.1. Furthermore, both Figure 1 and its caption have been revised to more clearly describe the physical system represented in the simulation.

Comments 4: It would be helpful to include setup photos and results images from the experiments. It is difficult to understand the experimental procedures through diagrams and charts alone.

Response 4: We agree and greatly appreciate your comment. To improve the understanding of the experimental procedure, we have included a photograph of the setup that was utilized in Figure 8, with the respective labels of the visible components.

Comments 5: Comparative analysis between actual experiment photos and data results would strengthen the paper. For instance, comparing images of force being applied to the sensor with how that force was recognized would help support the findings.

Response 5: While we agree that such a visual comparison between the application of forces to the surface of the sensor and the corresponding signal response would be beneficial, photographs of the procedures that lead to the responses displayed in the manuscript were not captured at the time that they happened. We believe that the inclusion of the photograph of the experimental setup already allows for a better understanding of the experimental procedure. Furthermore, we have included a photograph of the fully integrated device in Figure 6. However, if the reviewer considers that a better illustration is required, new photographs could be captured, recreating the procedure that took place at the time of the acquisition of the displayed data.

Round 2

Reviewer 1 Report

Comments and Suggestions for Authors

nothing

Author Response

The reviewer has not provided any further comments.